# Improving *Pinus densata* Carbon Stock Estimations through Remote Sensing in Shangri-La: A Nonlinear Mixed-Effects Model Integrating Soil Thickness and Topographic Variables

Dongyang Han [1], Jialong Zhang [2] , Dongfan Xu [3], Yi Liao [4], Rui Bao [5], Shuxian Wang [6] and Shaozhi Chen [7,*]

[1]  Research Institute of Forestry Policy and Information, Chinese Academy of Forestry, Beijing 100091, China; handy0519@caf.ac.cn
[2]  Forestry College, Southwest Forestry University, Kunming 650224, China; jialongzhang@swfu.edu.cn
[3]  Ministry of Education Key Laboratory for Biodiversity Science and Ecological Engineering, National Observations and Research Station for Wetland Ecosystems of the Yangtze Estuary, Shanghai Institute of EcoChongming (SIEC), Fudan University, Shanghai 200433, China; dfxu22@m.fudan.edu.cn
[4]  College of Mechanical and Electronic Engineering, Northwest Agriculture and Forestry University, Xianyang 712100, China; ianliao@nwafu.edu.cn
[5]  Institute of Southwest Survey and Planning, National Forestry and Grassland Administration, Kunming 650021, China; rui787185290@live.cn
[6]  Remote Sensing Center of Yunnan Province, Kunming 650034, China; esther@swfu.edu.cn
[7]  Chinese Academy of Forestry, Beijing 100091, China
*  Correspondence: szchen@caf.ac.cn; Tel.: +86-136-0122-3163

**Abstract:** Forest carbon sinks are vital in mitigating climate change, making it crucial to have highly accurate estimates of forest carbon stocks. A method that accounts for the spatial characteristics of inventory samples is necessary for the long-term estimation of above-ground forest carbon stocks due to the spatial heterogeneity of bottom-up methods. In this study, we developed a method for analyzing space-sensing data that estimates and predicts long time series of forest carbon stock changes in an alpine region by considering the sample's spatial characteristics. We employed a nonlinear mixed-effects model and improved the model's accuracy by considering both static and dynamic aspects. We utilized ground sample point data from the National Forest Inventory (NFI) taken every five years, including tree and soil information. Additionally, we extracted spectral and texture information from Landsat and combined it with DEM data to obtain topographic information for the sample plots. Using static data and change data at various annual intervals, we built estimation models. We tested three non-parametric models (Random Forest, Gradient-Boosted Regression Tree, and K-Nearest Neighbor) and two parametric models (linear mixed-effects and non-linear mixed-effects) and selected the most accurate model to estimate *Pinus densata*'s above-ground carbon stock. The results showed the following: (1) The texture information had a significant correlation with static and dynamic above-ground carbon stock changes. The highest correlation was for large-window mean, entropy, and variance. (2) The dynamic above-ground carbon stock model outperformed the static model. Additionally, the dynamic non-parametric models and parametric models experienced improvements in prediction accuracy. (3) In the multilevel nonlinear mixed-effects models, the highest accuracy was achieved with fixed effects for aspect and two-level nested random effects for the soil and elevation categories. (4) This study found that *Pinus densata*'s above-ground carbon stock in Shangri-La followed a decreasing, and then, increasing trend from 1987 to 2017. The mean carbon density increased overall, from 19.575 t·hm$^{-2}$ to 25.313 t·hm$^{-2}$. We concluded that a dynamic model based on variability accurately reflects *Pinus densata*'s above-ground carbon stock changes over time. Our approach can enhance time-series estimates of above-ground carbon stocks, particularly in complex topographies, by incorporating topographic factors and soil thickness into mixed-effects models.

**Keywords:** Landsat; *Pinus densata*; topographic information; soil thickness; multilevel nonlinear mixed-effects model

## 1. Introduction

Addressing climate change has emerged as a universal goal for nations worldwide in the 21st century [1]. Forests are pivotal to terrestrial ecosystems, boasting significant carbon sequestration capabilities [2]. Consequently, the Nature-Based Solutions (NbS) framework emphasizes forest resources as a key strategy in fighting climate change [3], providing myriad benefits to human society, such as ecological improvements [4]. Forest carbon stocks, which reflect the ability of forests to capture carbon to some extent, necessitate precise estimation [5]. Furthermore, evaluating changes in forest carbon stocks can disclose the direct effects of land use alterations on forest carbon reserves. These data foster a more evidence-based strategy for sustainable forest management, facilitating a preemptive approach to the challenges of climate change [6,7]. However, the extent of their impact greatly relies on how accurately we can measure and monitor these forest carbon stocks. As we contend with the escalating effects of climate change, the demand for accurate, scalable, and uniform methods to assess forest carbon reserves grows increasingly urgent.

Changes in forest carbon stocks on a regional scale are typically assessed through an inventory method. However, sites chosen through systematic sampling exhibit significant variability, making it challenging to represent the full distribution of a specific tree species across a region. This leads to spatial heterogeneity in carbon stock estimations using the inventory method [8]. Remote sensing estimation can mitigate spatial heterogeneity to a certain degree. The primary remote sensing models include single-year static data modeling methods and time-series data-based modeling methods [9–11]. While single-date static data modeling methods are relatively well developed, they struggle to capture changes in forest carbon stocks at regional metric scales. Consequently, models that treat biomass as a static variable, based on optical remote sensing data, face significant limitations [12]. Investigating factors that influence carbon stock changes and modeling future trends through dynamic biomass estimation using multi-period data is garnering increased interest. This process involves using temporal trend analysis to compare differences in the same spatial pixel values over multiple years, reflecting deforestation and forest biomass recovery [13,14]. Changes in spectral information values closely correlate with forest carbon stocks [15]. Utilizing the high temporal resolution of remotely sensed data's temporal spectral trajectories enables retrospective forest carbon stock estimation [16]. Remote sensing satellites' repeated sampling can align with NFI data on the same timescale [17]. Time trend analysis currently encompasses two main approaches: correlating time-series data from sample points with pixel data, and correlating changes over time in sample points with changes in pixel values. Landsat time-series estimations of single-date models have proven capable of characterizing forest disturbance and recovery and improving above-ground biomass (AGB) estimation accuracy [18], primarily using the LandTrendr algorithm for analysis [19]. Changes in remote sensing spectral features correlate with changes in forest biomass, reflecting the trend of biomass changes to a certain extent [20]. Using "dynamic" variables, such as the amount of change in pixels, may be more effective for estimating carbon stock changes in repeated measurements. Puliti et al. constructed a time series combining national forest inventory data with remotely sensed data, demonstrating the applicability of the amount-of-change model to repeated measurements and the improvement in Landsat time-series data over previous data sets [21]. However, remote sensing estimation models still entail significant uncertainty, and the accuracy of dynamic variables depends on the availability of repeated measurement data and the model's interpretability [22].

Estimation models can be categorized into parametric and non-parametric models. Parametric models, including multiple regression models, may be linear [23] or nonlinear [24]. However, traditional multiple regression models often show low estimation accuracy [25] and do not account for the spatial heterogeneity of forests. In contrast, non-parametric models, while often providing superior accuracy, tend to suffer from limited portability across different contexts. National forest inventory data, measured at regular intervals, offer limited temporal coverage, making them inadequate for tracking changes in forest carbon stocks during years without measurements. Non-parametric estimation

models, relying on fixed area or time fits, may lead to increased uncertainty when applied to remote sensing images from different areas or times [26,27]. Moreover, carbon stock data from different national forest inventory plots are non-independent data, influenced by a myriad of factors including topography and climate, which introduces a complex layer of spatial heterogeneity in the information gathered on forest stands [28–30]. Mixed-effects models, known for combining fixed- and random-effects parameters, effectively tackle data with hierarchical structures, clustering, non-independence, and non-constant variance [31]. Researchers frequently use mixed-effects models in studies that require repeated data observations, such as those in engineering and biology [32–34], especially when dealing with dimensionally rich data, such as forest data. Developers have created various mixed-effects models that integrate diverse forest information with topographic heterogeneity, crucial for understanding mountain forest growth [35]. The inclusion of topography as a random effect in the models has revealed how slope orientation differentially affects elevation and aspect, aiding in the prediction of vegetation species composition across elevation gradients [36]. Likewise, mixed-effects models incorporating topographic and climatic factors more effectively assess forest conditions, particularly at high elevations and in ecologically sensitive areas [37]. Furthermore, mixed-effects models adeptly unravel the impact of soil depth on the soil organic carbon content in forests, highlighting how topography influences carbon exchange between soil and trees [38]. Additionally, the use of mixed-effects models that include topographic factors has been instrumental in designing forest management plans at a regional scale [39]. In the analysis of forest productivity, species composition, and soil carbon cycling at high elevations with complex topography, topography–soil models provide enhanced flexibility and tend to surpass simple linear regression models by mitigating the linear bias of residuals, thereby more accurately estimating extreme values. Despite these models' success, further research is essential to explore the correlation between the choice of random effects in nonlinear mixed-effects models and remotely sensed spectral data, and to assess whether mixed-effects models relying on time-series data from remote sensing can more accurately estimate vegetation carbon stocks, particularly for long-term changes at the regional scale. Consequently, innovative approaches are crucial for overcoming the limitations and diminishing the uncertainty associated with the remote sensing time-series estimation of non-independent data.

In this study, we use mixed-effects modeling to investigate the relationship between soil and terrain variations and the spectral and textural information from remotely sensed imagery. We construct linear and nonlinear mixed-effects parameter models for two parameters, "static" values of carbon stock versus spectral and textural values, and "dynamic" values of carbon stock variation versus spectral and textural variation, using long time series of Landsat and NFI data, respectively. We estimate the dynamic changes in the forest carbon stocks of *Pinus densata* in Shangri-La, Yunnan Province, over a 30-year period using "static" values of carbon stocks versus spectral and textural values and "dynamic" values of carbon stock changes versus spectral and textural changes, respectively, introducing topographic and soil factors as mixed effects into the model. For comparison, we also develop three machine learning models to evaluate the effectiveness of the mixed-effects model. By elucidating the effects of topography and soil conditions on the uncertainty of remotely sensed long-term carbon stock estimation, we aim to provide a methodology for improving the parameterization of forest carbon stock estimation in regions with complex topography and ecological fragility. This study improves the accuracy of long-term remotely sensed carbon stock estimates.

## 2. Materials and Methods

### 2.1. Study Area

The study area, Shangri-La City in Yunnan Province, southwest China (99°20′–100°19′ E, 26°52′–28°52′ N), features a topography marked by lower elevations in the south and higher in the north, with obvious differences in elevation and distinct gradients and slope directions, and it is the hinterland of the Hengduan Mountains on the Qinghai–Tibet

Plateau. The region spans 11,613 square kilometers and boasts an average elevation of 3459 m. Shangri-La is abundant in forest resources, featuring a forest coverage rate of 75%.

Shangri-La's unique natural conditions foster a complex and diverse array of soil types, exhibiting a distinct vertical distribution pattern correlated with altitude. This pattern ranges from Mountain Base-Zone Soils at lower altitudes to Alpine Cold Desert Soils at the highest elevations: Mountain Base-Zone Soils (Yellow-Brown Soils) are found at altitudes of 2600–2900 m; Brown Soils are present at 2900–3300 m; Dark Brown Soils (including Gray-Brown Soils and Acidic Brown Soils) are located at 3200–3700 m; Dark Coniferous Forest Soils occur at 3500–4000 m; Alpine Meadow Soils span altitudes of 4000–4500 m; Cold Alpine Soils range from 4500 to 4800 m; Alpine Cold Desert Soils are found above 4800 m, marking the highest soil classification in this vertical distribution [40].

The *Pinus densata* sample sites in this study are predominantly situated within zones containing Yellow-Brown Soil, Dark Brown Soil, Red Soil, Black Limestone Soil, Brown Soil, Coarse-Boned Brown Soil, and Dark Red Soil. This diversity of soil types underlines the ecological complexity of Shangri-La, providing crucial context for our analysis of alpine pine distribution and its environmental interactions.

The predominant species found in the area include *Quercus semicarpifolia*, *Pinus yunnanensis*, *Pinus densata*, *Picea asperata*, and *Abies fabri*. *Pinus densata*, belonging to the Pinaceae family, is characterized by its tough wood, fine texture, high resin content, and deep roots. This light-loving species thrives on dry and infertile soils. Its primary distribution spans western Sichuan, southern Qinghai, eastern Tibet, and the northwestern Yunnan areas. Typically found at altitudes around 3000 m, *Pinus densata* commonly grows on sunny slopes and riverbanks, forming pure stands. Below elevations of 3000 m, these trees frequently intermix with *Yunnan pine*. Covering 1848.18 km$^2$, or 16.18% of the Shangri-La region's total land area, forests of *Pinus densata* play a vital role in regulating atmospheric carbon dioxide concentrations and in maintaining the balance between carbon and oxygen levels (Figure 1).

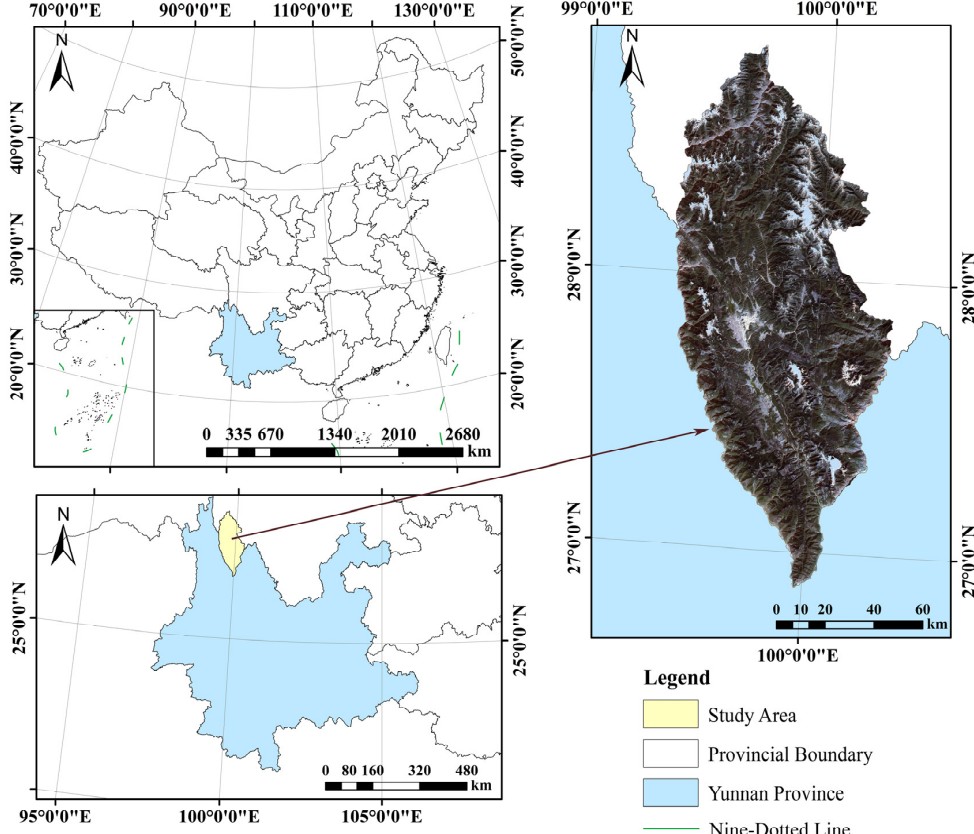

**Figure 1.** Overview of the study area.

### 2.2. Field and Remote Sensing Data

The data for the fixed sample sites in this study were sourced from China's Continuous National Forest Inventory, a survey conducted at five-year intervals. Starting in 1987 and concluding in 2017, the inventory was carried out seven times: 1987, 1992, 1997, 2002, 2007, 2012, and 2017. The collected data included the height and diameter at breast height of *Pinus densata* at 32 fixed sample sites, as well as information on the soil properties and soil thickness of the sample sites. The distribution ranges of *Pinus densata* for the specified years were derived from the sub-compartment data of *Pinus densata* within China's Forest Management Inventory.

Since some fixed sample sites lacked seven consecutive replications, this study selected 20 fixed sample sites with seven consecutive replications as baseline data (Figure 2). The re-mote sensing time-series image data include Landsat 5 TM and atmospherically corrected Landsat 8 OLI surface re-reflection archive data, provided by the National Aeronautics and Space Administration (NASA) and the United States Geological Survey (USGS). The three images with the lowest cloud cover were chosen per corresponding year, resulting in a total of 18 Landsat 5 TM and three Landsat 8 OLI images. The cloud coverage ranged from a low of 0.13% to a high of 23.89% (Table 1).

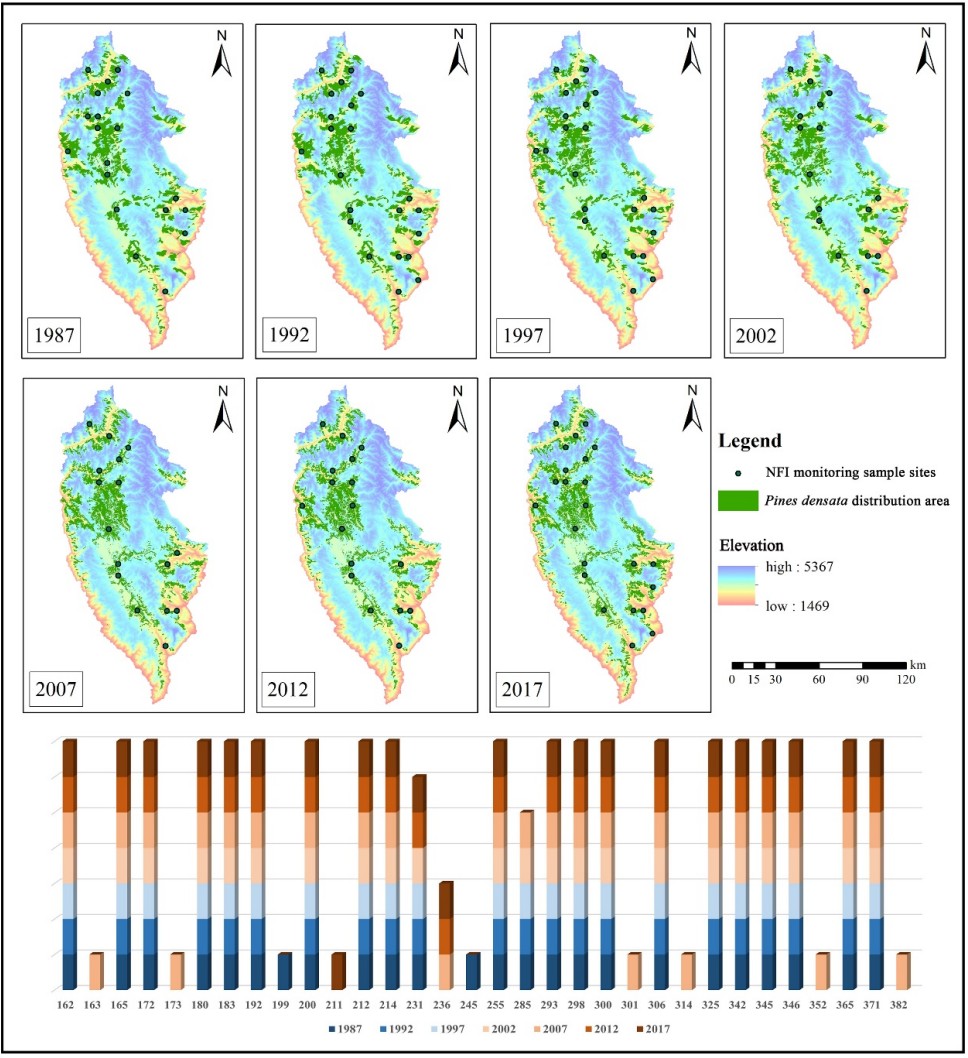

**Figure 2.** Distribution area of *Pinus densata* from 1987 to 2017; 162–382 are the NFI fixed sample plot numbers.

**Table 1.** Basic information of Landsat time-series images in the study area.

| Sensor Type | Survey Year | Strip Number | Line Number | Acquisition Time | Cloud Cover/(%) |
|---|---|---|---|---|---|
| Landsat5 TM | 1987 | 131 | 041 | 23 December 1987 | 1.00 |
| | | 132 | 041 | 30 December 1987 | 10.60 |
| | | 132 | 040 | 30 December 1987 | 23.89 |
| | 1992 | 131 | 040 | 16 November 1991 | 8.05 |
| | | 132 | 041 | 7 November 1991 | 2.44 |
| | | 132 | 041 | 7 November 1991 | 5.24 |
| | 1997 | 131 | 041 | 16 November 1997 | 7.00 |
| | | 132 | 040 | 6 October 1997 | 16.00 |
| | | 132 | 041 | 7 November 1997 | 4.00 |
| | 2002 | 131 | 041 | 29 October 2002 | 0.13 |
| | | 132 | 040 | 5 January 2002 | 0.15 |
| | | 132 | 041 | 5 January 2002 | 2.80 |
| | 2007 | 131 | 041 | 1 March 2007 | 1.00 |
| | | 132 | 040 | 3 January 2007 | 23.00 |
| | | 132 | 041 | 15 October 2006 | 15.00 |
| | 2012 | 132 | 041 | 13 October 2011 | 19.52 |
| | | 132 | 040 | 14 January 2011 | 18.15 |
| | | 131 | 041 | 7 January 2011 | 0.22 |
| Landsat 8 OLI | 2017 | 132 | 040 | 16 December 2017 | 0.78 |
| | | 132 | 041 | 16 December 2017 | 0.45 |
| | | 131 | 041 | 25 December 2017 | 0.30 |

In this study, we constructed linear and nonlinear mixed-effects models to investigate whether incorporating topographic and soil constraints into a parametric approach for remote sensing can improve the estimation of above-ground forest carbon stocks over long time series. We used Landsat 5 Thematic Mapper (TM) and Landsat 8 Operational Land Imager (OLI) time-series imagery and *Pinus densata* NFI plots in Shangri-La, Yunnan, for our data. We then compared the performance of these models with three non-parametric models. The models were developed using Python 3.7.4, Matlab R2021b, and STATAMP 17 software (Figure 3).

*2.3. Basic Data Processing*

First, the DN values were converted to radiometric values using the Radiometric Correction Tool [41]. Then, atmospheric correction was performed with the Fast Line of Sight Atmospheric Analysis of Spectral Hypercubes (FLAASH) module [42]. The calibrated SPOT-5 image was used as a reference for correcting the image coordinate system to Beijing 1954. The binomial correction method was employed, with at least 30 ground control points selected per scene image. The SPOT-5 image was resampled to a resolution of 30 m × 30 m through bilinear interpolation, ensuring the calibration error of the two images was controlled within one pixel. Topographic correction was carried out using the slope-matching method. After the secondary correction [43], the mean values of topographic shading on the north and south slopes became similar. The reflectance values of the shaded parts of the slopes were compensated by those of the sunlit parts (Figure 4). Finally, the images were stitched together.

To obtain the most relevant modeling factors for above-ground forest carbon stock, we extracted vegetation indices, ratio factors, texture factors, and information enhancement factors based on the single-band factors (Table 2).

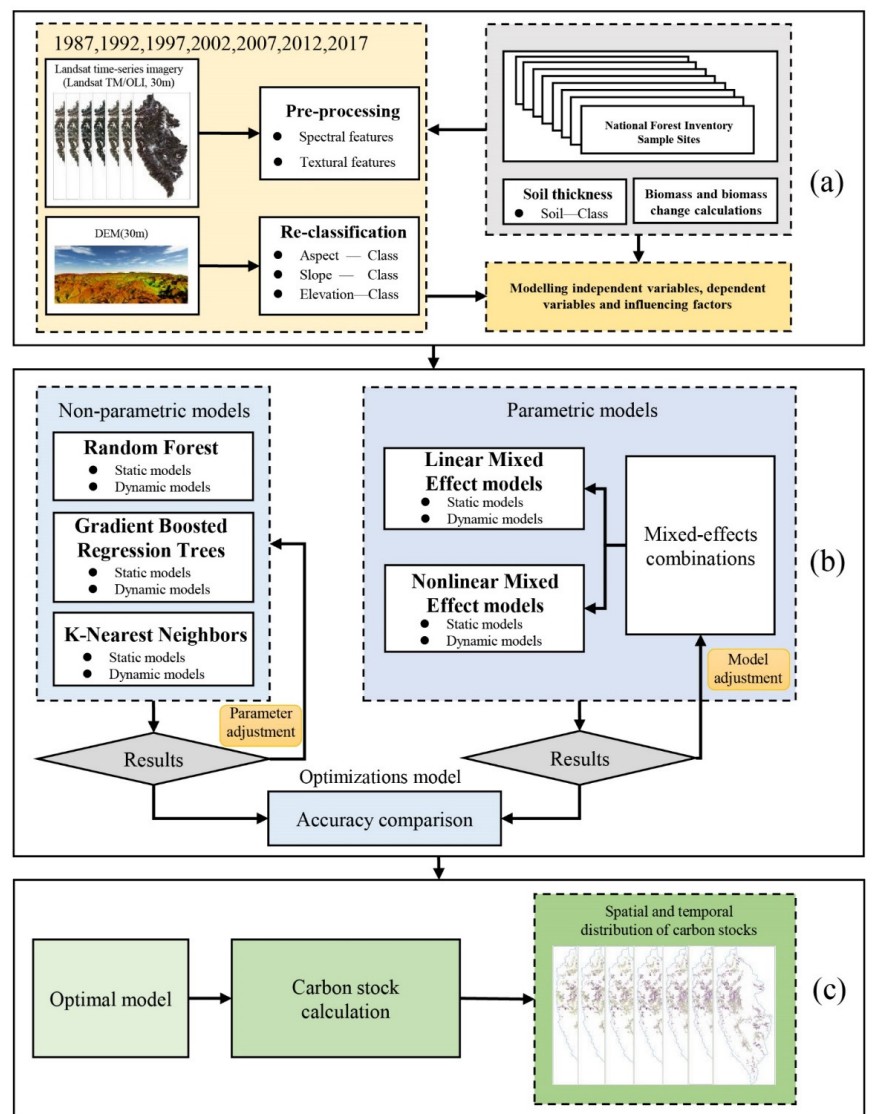

**Figure 3.** Research flowchart. The process involved three main steps: (**a**) acquiring and processing time-series remote sensing data and NFI data, followed by selecting, extracting, and evaluating remote sensing variables; (**b**) creating models and assessing their accuracy; and (**c**) applying optimal modeling approaches to develop spatial and temporal maps of above-ground carbon stocks.

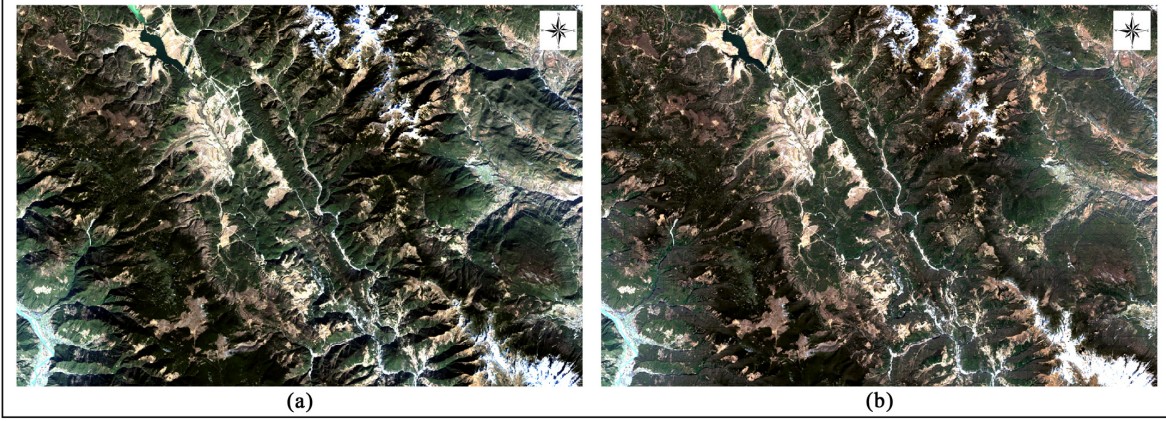

**Figure 4.** Comparison of image preprocessing results: (**a**) unprocessed original image detail; (**b**) pre-processed image detail.

**Table 2.** Remote sensing eigenvalue variables.

| Factor Type | Factor |
| --- | --- |
| Vegetation indices | ND54, ND64, NDVI, ND53, ND65, ND32, RVI, DVI |
| Image enhancement factor | PCA |
| Original band factor | B1, B2, B3, B4, B5, B6, B7 |
| Simple band ratio factor | B4/Albedo, (B5 × B4)/B7 |
| Texture information factor | Mean (Me), variance (VA), homogeneity (HO), contrast (CO), dissimilarity (DS), entropy (EN), second moment (SM), correlation (CC), skewness (SK) |

(1) Landsat 5 Thematic Mapper (TM): B1 = Band 1 Visible Blue (0.45–0.52 μm) 30 m, B2 = Band 2 Visible Green (0.52–0.60 μm) 30 m, B3 = Band 3 Visible Red (0.63–0.69 μm) 30 m, B4 = Band 4 Near-Infrared (0.76–0.90 μm) 30 m, B5 = Band 5 Near-Infrared (1.55–1.75 μm) 30 m, B6 = Band 6 Thermal (10.40–12.50 μm) 120 m, B7 = Band 7 Mid-Infrared (2.08–2.35 μm) 30 m; (2) Landsat 8 Operational Land Imager (OLI): B1 = Band 1 Coastal Aerosol (0.43–0.45 μm) 30 m, B2 = Band 2 Blue (0.450–0.51 μm) 30 m, B3 = Band 3 Green (0.53–0.59 μm) 30 m, B4 = Band 4 Red (0.64–0.67 μm) 30 m, B5 = Band 5 Near-Infrared (0.85–0.88 μm) 30 m, B6 = Band 6 SWIR 1 (1.57–1.65 μm) 30 m, B7 = Band 7 SWIR 2 (2.11–2.29 μm) 30 m; (3) PCA = Principal Component Analysis; (4) ND54 = (B5 − B4)/(B5 + B4), ND64 = (B6 − B4)/(B6 + B4), ND53 = (B5 − B3)/(B5 + B3), ND65 = (B6 − B5)/(B6 + B5), ND32 = (B3 − B2)/(B3 + B2), NDVI = (B4 − B3)/(B4 + B3), RVI = B4/B3, DVI = B4 − B3; (5) Albedo = B1 + B2 + B3 + B4 + B5 + B7. (6) R is for Extraction Window: 3 × 3, …, 19 × 19; total of 9 odd-sized windows.

The above-ground biomass of *Pinus densata* sample plots in the inventory of forest resources was calculated according to the formula for the single-wood biomass of *Pinus densata* obtained from the study [44], and the single-wood biomass model is shown in Equation (1),

$$AGB = 0.073 \times DBH^{1.739} \times H^{0.880} \tag{1}$$

where *AGB* is the is the above-ground biomass of a single wood (kg), DBH is the diameter at breast height (cm), and *H* is the hHeight. An approximate estimation of sample plot biomass was made based on mean diameter at breast height, mean tree height, and number of plants.

Upon completion of the biomass calculations for *Pinus densat*, carbon conversion factors were used to determine the carbon stocks of the species. The carbon conversion factors for *Pinus densat* were obtained from the *Guidelines for Accounting for Forest Ecosystem Carbon Stocks*, published by the State Forestry and Grassland Administration of China. Additionally, the biomass expansion factors (BEFs) and stem volume densities (SVDs) for *Pinus densata* were provided for researchers' reference [45] (Table 3).

**Table 3.** Biomass conversion factor.

| Tree Type | SVD | BEF | CF |
| --- | --- | --- | --- |
| *Pinus densata* | 0.413 | 1.6509 | 0.501 |

*Pinus densata* carbon stock was calculated as shown in Equation (2),

$$C = AGB \times 0.501 \tag{2}$$

where *C* is the forest carbon stock, *AGB* is the above-ground biomass, and 0.501 is the carbon conversion factor.

Existing studies have shown that variation models are superior to static data models in terms of estimation accuracy and stability [16,20,21,46]. Therefore, in this study, the amount of change in *Pinus densata* carbon stock was calculated by calculating the carbon stock of *Pinus densata* in each of the sample plots of NFI. Six categories of change intervals were used, namely 5-year intervals (1987–1992, 1992–1997, 1997–2002, 2002–2007, 2007–2012, 2012–2017), 10-year intervals (1987–1997, 1992–2002, 1997–2007, 2002–2012, 2007–2017), 15-year intervals (1987–2002, 1992–2007, 1997–2012, 2002–2017), 20-year intervals (1987–2007, 1992–2012, 1997–2017), 25-year intervals (1987–2012, 1992–2017), and a

30-year period (1987–2017). The formula for calculating the amount of carbon stock change is shown in Equation (3),

$$\Delta AGCS = AGCS_a - AGCS_b \tag{3}$$

where $\Delta AGCS$ is the amount of above-ground carbon stock change in *Pinus densata*, and $AGCS_a$ and $AGCS_b$ are the above-ground carbon stock of *Pinus densata* in two different years.

The amount of change in the remote sensing factor needed for modeling was calculated using the formula shown in Equation (4),

$$\Delta RSF = RSF_a - RSF_b \tag{4}$$

where $\Delta RSF$ is remote sensing factor variation, and $RSF_a$ and $RSF_b$ are remote sensing factor values for two different years.

A remote sensing estimation model was constructed using the continuous inventory sample plot data of forest resources. To distinguish the accuracy of the estimation models for multi-period continuous carbon stock data and carbon stock change data, remote sensing estimation models were constructed using the AGCS and AGCS changes in the continuous inventory sample plots as dependent variables. These variables were correlated with remote sensing eigenvalues and eigenvalue change, respectively. These models are referred to in Section 3 as static and dynamic models, respectively. Table 4 displays the carbon stock calculations and the resulting statistics for the national forest inventory sample plots. The forest inventory data included 117 valid sample plots from 1987 to 2017, and the above-ground carbon stock data, calculated from these, were subsequently used to determine carbon stock changes at 5-, 10-, 15-, 20-, 25-, and 30-year intervals. A total of 263 sets of change data were computed, comprising 93 sets for the 5 years, 70 sets for the 10 years, 50 sets for the 15 years, 41 sets for the 20 years, 27 sets for the 25 years, and 15 sets for the 30 years. Out of the 117 sets of carbon stock data, 82 sets of sample sites were randomly selected as training data for the static model, and the remaining 35 sets of sample sites were utilized as test data for the static model. Of the carbon stock change data, 183 data sets were randomly selected as training data for the dynamic model, while the remaining 80 data sets were employed as test data for the dynamic model. These models are referred to as static and dynamic models, respectively.

**Table 4.** Statistical results of carbon stock and carbon stock variation.

| Carbon Stock Type | Statistics | Training Data 70% | | | Test Data 30% | | |
|---|---|---|---|---|---|---|---|
| | | Average Tree Height/m | Mean DBH/cm | Carbon Stock (t·hm⁻²) | Average Tree Height/m | Mean DBH/cm | Carbon Stock (t·hm⁻²) |
| AGCS | Mean | 10.54 | 18.93 | 27.10 | 10.15 | 14.78 | 31.88 |
| | Max | 12.5 | 91.7 | 85.64 | 17.0 | 21.5 | 74.62 |
| | Min | 3.4 | 6.0 | 1.03 | 3.1 | 6.1 | 1.78 |
| | Stand error | 4.91 | 15.86 | 19.06 | 3.95 | 6.77 | 18.55 |
| Amount of AGCS change | Mean | 0.84 | −0.76 | 7.37 | 2.14 | 0.493 | 15.80 |
| | Max | 6.3 | 16.34 | 51.52 | 9.3 | 16.64 | 56.13 |
| | Min | −12.4 | −67.4 | −36.89 | −12.6 | −65.9 | −33.05 |
| | Stand error | 2.90 | 10.71 | 14.04 | 4.50 | 12.12 | 20.56 |

*2.4. Soil and Terrain Factor Reclassification*

We classified aspect, slope, and elevation within the forest inventory sample plot data. Aspect was classified into nine grades according to the standardized criteria. The grading was based on the difference in the main distribution range of slope, and the maximum value of the slope in the forest inventory sample plots was 54.47°, and the minimum value was 9.32°. Therefore, the slopes were categorized into eight classes: 0–10°, 10–15°, 15–20°, 20–25°, 25–30°, 30–35°, 35–40°, and 40–55°. The maximum value of the elevation distribution of the forest inventory sample plot data was 3873 m, and the minimum value was 2715 m. Therefore, the elevation was divided into six classes of 2500~3000, 3000~3200,

3200~3400, 3400~3600, 3600~3800, and 3800~3900 m (Figure 5). Soil thickness was included in the forest inventory sample data, and the maximum value of soil thickness was 90 cm and the minimum value was 0 cm, and the remeasured soil thickness varied from year to year. In order to differentiate the effect of soil thickness on biomass estimation on different terrains, the soil depth as a modeling factor was graded as 0–10, 10–30, 30–40, 40–50, 50–60, 60–70, 70–80, and 80–90 cm in eight classes (Table 5).

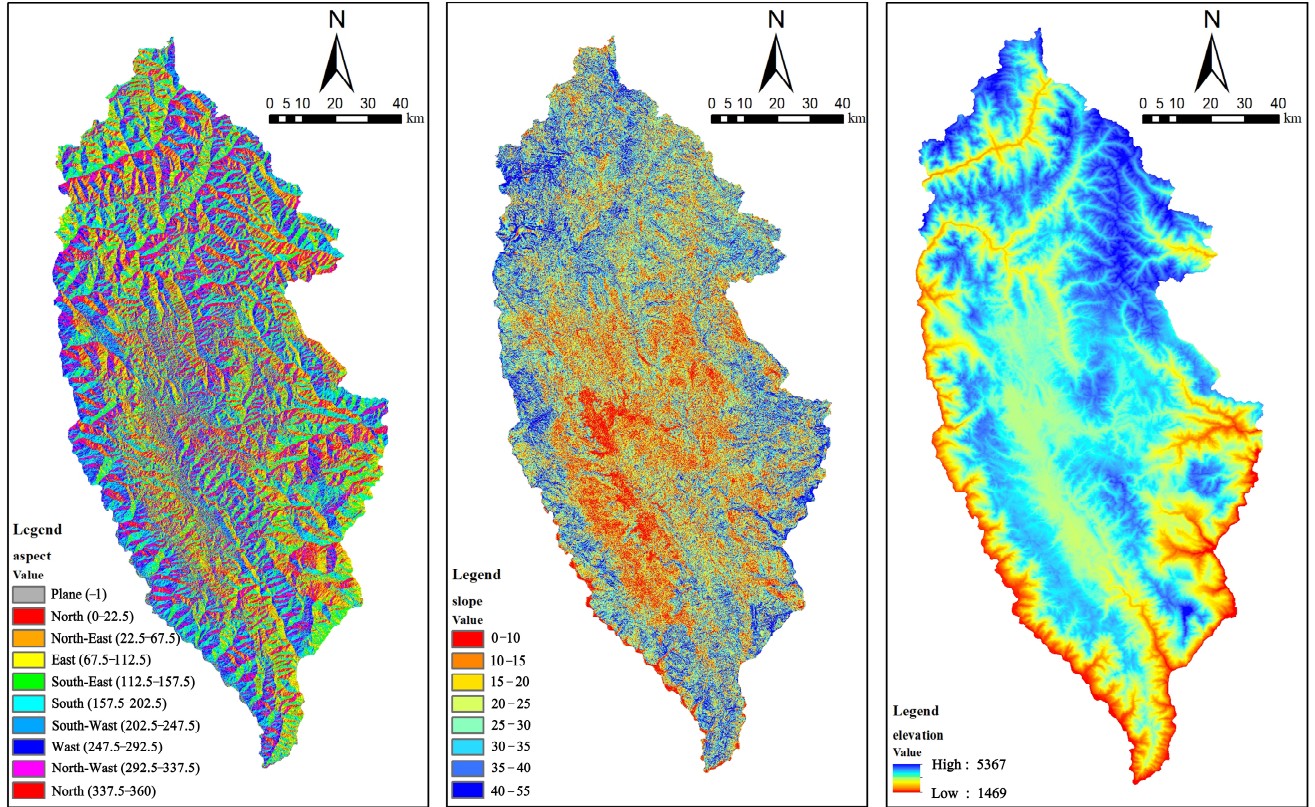

**Figure 5.** Elevation, aspect, and slope-class map.

**Table 5.** Aspect, slope, elevation, and soil classes.

| Aspect Class | | Slope Class | | Elevation Class | | Soil Class | |
|---|---|---|---|---|---|---|---|
| Class | Aspect Bearing | Class | Slope Range/(°) | Class | Elevation Range/m | Class | Soil Thickness/cm |
| 1 | Plane | 1 | 0~10 | 1 | 2500~3000 | 1 | 0~10 |
| 2 | North | 2 | 10~15 | 2 | 3000~3200 | 2 | 10~30 |
| 3 | North–East | 3 | 15~20 | 3 | 3200~3400 | 3 | 30~40 |
| 4 | East | 4 | 20~25 | 4 | 3400~3600 | 4 | 40~50 |
| 5 | South–East | 5 | 25~30 | 5 | 3600~3800 | 5 | 50~60 |
| 6 | South | 6 | 30~35 | 6 | 3800~3900 | 6 | 60~70 |
| 7 | South–West | 7 | 35~40 | | | 7 | 70~80 |
| 8 | West | 8 | 40~55 | | | 8 | 80~90 |
| 9 | North–West | | | | | | |

### 2.5. Carbon Stock Estimation Model Construction

2.5.1. Correlation Analysis of Remote Sensing Information

A total of 577 remotely sensed variables and their corresponding changes were extracted and analyzed to evaluate their correlation with changes in carbon stocks and carbon storage on the ground (Figure 6). The top 10 variables demonstrating the strongest correlations were selected to construct linear stepwise regression models. To avoid over-fitting and co-linearity risks, several variables were excluded from the models, and the

independent variables were deterministically modeled based on the stepwise regression outcomes [47]. For static modeling, R19B4SM and R01B5CT were identified as modeling factors, employing AGCS as the dependent variable and remotely sensed characteristics as the independent variables. In the case of the dynamic model, R19B5EN, R15B5EN, R17B5ME, R15B5ME, R19B5ME, and R17B5DS were recognized as modeling factors, using changes in AGCSs as the dependent variable and alterations in remotely sensed characteristics as the independent variables.

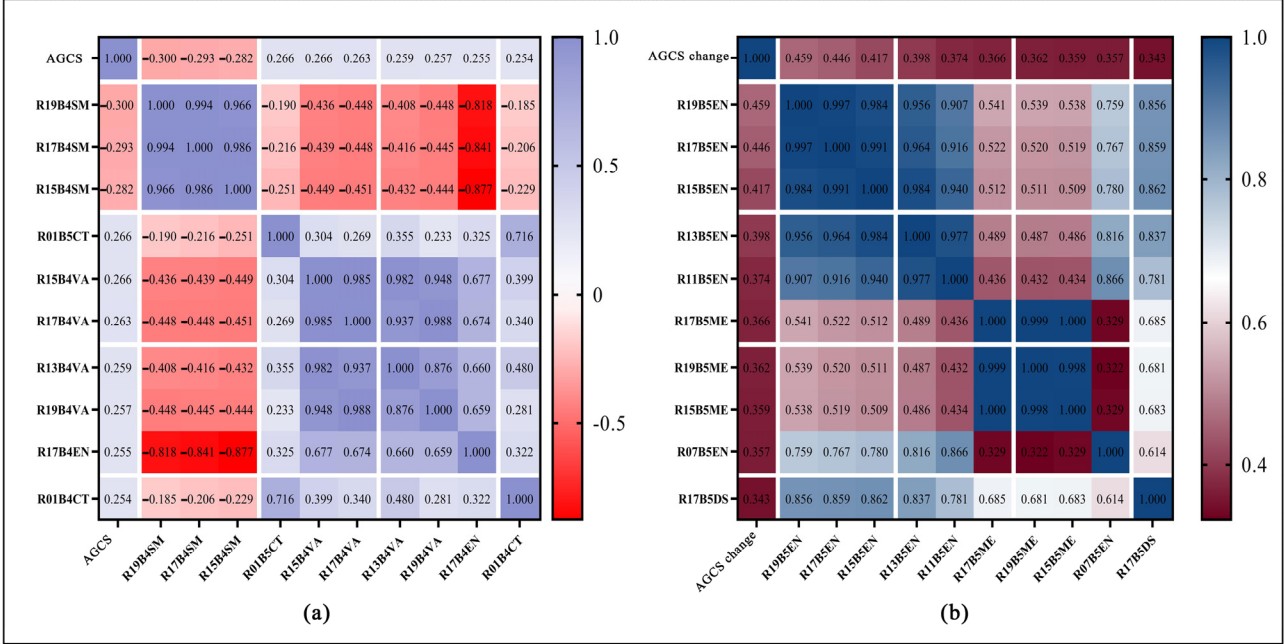

**Figure 6.** Modeling variable correlation analysis: (**a**) heat map for correlation analysis of static AGCS data and remotely sensed information; (**b**) heat map for correlation analysis of dynamic AGCS change data and the amount of change in remotely sensed information.

### 2.5.2. Non-Parametric Model Construction

Non-parametric models are known for achieving higher fitting and prediction accuracy compared to traditional parametric models [48]. To distinguish the difference in prediction accuracy between non-parametric models and mixed-effects models, this study constructed three static and dynamic non-parametric models: the Random Forest (RF) model, Gradient-Boosted Regression Tree (GBRT) model, and K-Nearest Neighbor (K-NN) model.

A Random Forest (RF) model consists of multiple decision trees where each tree is trained independently, reducing the correlation between them. As the number of decision trees increases, the generalization error converges to a limit [49], and the final result obtained is the average of all decision tree predictions. Due to this, stochastic forests are widely used for estimating forest biomass or carbon stocks [50–52].

The Gradient-Boosted Regression Tree (GBRT) model relies on a boosting algorithm to train multiple decision tree models. At each iteration, the data set samples are weighted according to the results from the previous step, and then, iterated again to continuously optimize the residuals and improve the performance of the decision tree [53]. The GBRT model can automatically calculate the importance of variables [54], with related techniques found in [55]. Additionally, the GBRT algorithm combats the overfitting of new data and delivers better results in estimating time scales [56].

The K-Nearest Neighbor (K-NN) strategy is widely utilized in measuring and monitoring forest biomass and carbon stocks [57,58]. It offers significant advantages in forest inventory applications, such as estimating missing values in forest inventory and monitoring data [59,60], the thematic mapping of different forest types [61,62], and small-scale

estimation [63]. The K-NN algorithm is employed to solve classification problems by determining a sample's class based on the proximity and similarity of neighboring points within a range of K. Although the K-NN operating principle is simple and the algorithm is tractable, it can be slow when dealing with large training data sets.

All three models were implemented using the Python 3.7.4 language environment and the scikit-learn machine learning tool. The specific parameter settings for each of the three machine learning models are presented in the main results section.

### 2.5.3. Parametric Model Construction

The construction of a mixed-effects model necessitates the separate identification of fixed and random effects. Our extraction results for topographic and soil data revealed that both the topography and soil thickness of the sample sites varied significantly over 30 years when measured every five years. These topographic differences in the *Pinus densata* sample sites in this study might contribute to errors in carbon stock estimation resulting from differences in soil thickness and topographic variability [64]. As a consequence, we constructed linear and nonlinear mixed-effects models by considering reclassified soil thickness as a random effect. Topographic factors were treated as fixed and random effects, respectively.

Linear mixed-effects models (LMEMs), also referred to as multilevel or hierarchical models, incorporate a random-effects parameter, unlike multiple linear regression [65]. These models can be applied to a wide array of data types and offer increased flexibility and effectiveness when handling non-normally distributed data [66,67]. The fundamental expression form of the linear mixed-effects model is illustrated in Equation (5). In this study, a linear mixed-effects model was fitted using the lme module of Matlab2021 software, and soil class, slope class, slope class, and elevation class were modeled as random effects and fixed effects, respectively, in the model, constructed using continuous inventory data.

$$\begin{cases} AGCS_{ij} = \beta_{0ij} + \beta_{1ij}RSF_{ij} + e_{ij} \\ \quad e_{ij} \sim N(0, \sigma_e^2 I) \end{cases} \tag{5}$$

In this model, $AGCS_{ij}$ and $RSF_{ij}$ are the values of the dependent and independent variables for the *j*th repeated observation of the *i*th study subject; $\beta_{0ij}$ is a P-vector with fixed overall parameters, $\beta_{1ij}$ is the vector of random effects q for study individual *i*; and $e_{ij}$ is an independently distributed noise term (random error term).

In the second step, we selected the underlying nonlinear functions. We chose a total of eight commonly used nonlinear functions and stand growth functions with specific forest growth characteristics [68] as candidate underlying models (Table 6).

**Table 6.** Nonlinear basic model.

| Function Number | Function Expression | Function Name |
|:---:|:---:|:---:|
| 1 | $Y = \beta_0 x^{\beta_1}$ | Power function |
| 2 | $Y = \beta_0 + \beta_1 In(x)$ | Natural logarithmic function |
| 3 | $Y = \beta_0 + \beta_1 x^2 + \beta_2 x^3$ | Polynomial functions |
| 4 | $Y = e^{(\beta_0 + \beta_1 x)}$ | Growth function |
| 5 | $Y = \beta_0 e^{\beta_1 x}$ | Natural exponential function |
| 6 | $Y = \beta_0 + \beta_1/x$ | Hyperbolic function |
| 7 | $Y = e^{\beta_0 + \beta_1/x}$ | S-shaped curve function |
| 8 | $Y = \frac{1}{1/u + \beta_0(\beta_1^x)}$ | Logistic Stix equation |
| 9 | $Y = \frac{\beta_0}{1 + \beta_1 e^{-\beta_2 x}}$ | Forest stand growth Model |

We subsequently concentrated on constructing nonlinear mixed-effects models (NLMEMs) using remotely sensed eigenvalues as modeling factors. These models were divided into single-level NLMEMs and multilevel NLMEMs [69]. The basic expressions for single-level NLMEMs are depicted in Equation (6).

$$\begin{cases} AGCS_{ij} = f\left(\phi_{ij}, v_{ij}\right) + e_{ij}, i = 1, \cdots, M, j = 1, \cdots, n_i \\ \phi_{ij} = A_{ij}\beta + B_{ij}b_i \\ b_i \sim N\left(0, \sigma^2 D\right) \\ e_{ij} \sim N\left(0, \sigma^2 R_i\right) \end{cases} \qquad (6)$$

where $AGCS_{ij}$ and $v_{ij}$ are the values of the dependent and independent variables for the $j$th repeated observation of the $i$th study subject, respectively; $M$ is the number of study subjects; $n_i$ is the number of repeated observations for the $i$th study subject; $e_{ij}$ is an independently distributed noise term (random error term); $f$ is a nonlinear function of the prediction vector and the parameter vector; $\varphi_{ij}$ is a vector of formal parameters appearing in nonlinear form in the function f; $\beta$ is a p-vector with fixed overall parameters; $b_i$ is the vector of random effects q associated with individual $i$; the matrices $A_i$ and $B_i$ are design matrices of fixed and random effects of size r × p and r × q; and $\sigma^2 D$ is the covariance matrix.

Upon determining the nature of the underlying nonlinear function, we constructed a multilevel mixed-effects model. Multilevel mixed-effects models (MMEMs) account for the autocorrelation of random factors between plots and within the same plot, providing a better fit for the data than multiple linear regression models and single-level mixed-effects models. These models have been widely used in agriculture, forestry, and medicine [70–72]. The continuous forest inventory sample plots utilized in this study were fixed, featuring different stand conditions and a substantial timespan. Therefore, strong autocorrelation and heteroskedasticity characteristics appeared among sample plots. Some research has demonstrated that multilevel nonlinear mixed-effects models significantly outperform traditional models in eliminating sample autocorrelation and heteroskedasticity [73]. In this study, using the StataMP 16.0 multilevel mixed-effects model module, we added soil thickness as a random-effects factor to the nonlinear mixed-effects model based on a single-level model constructed from single-period data. We considered samples with varying aspects (first level), slopes (second level), and elevations (third level). Soil thickness served as the primary random effect, while the three topographic factors of aspect, slope, and elevation were combined with a soil factor for multilevel random-effects nesting (Table 7). The expressions for the models are provided in Equation (7),

$$\begin{cases} AGCS_{ijk} = f\left(\phi_{ijk}, v_{ijk}\right) + e_{ijk}, i = 1, \cdots, M, j = 1, \cdots, M_i, \quad k = 1, \cdots, n_{ij}, \\ \phi_{ijk} = A_{ijk}\beta + B_{i,jk}b_i + B_{ijk}u_{ij} \\ b_i \sim N(0, D_1) \\ b_{ij} \sim N(0, D_2) \\ e_{ij} \sim N\left(0, \sigma^2 R_{ij}\right) \end{cases} \qquad (7)$$

where $AGCS_{ijk}$ is the kth observation within the $j$th 2nd level at the $i$-th 1st level; $M$ is the number of subgroups at the 1st level; $M_i$ is the number of subgroups at the 2nd level within the 1st level; $n_{ij}$ is the number of observations within the $j$th 2nd level at the $i$-th level; $f$ is a function containing a parameter vector $\varphi_{ijk}$ and a covariance vector $v_{ijk}$; $\beta$ is a (p × 1)-dimensional fixed-effects vector; $A_{ijk}$ is the design matrix; $b_i$ is the (q1 × 1)-dimensional random-effects vector with a variance–covariance matrix $D_1$ at level 1; $b_{ij}$ is a (q2 × 1)-dimensional random-effects vector with a variance–covariance matrix $D_2$ at level 2; $b_i$ and $b_{ij}$ are uncorrelated; $b_{i,jk}$ and $b_{ijk}$ are the design matrices of random effects; $e_{ij}$ is the error term obeying normal distribution; $\sigma^2$ is the variance; and $R_{ij}$ is the variance–covariance matrix within the $j$th 2nd level of the ith 1st level [74].

We compare the goodness of fit of the mixed-effects model using three metrics: log likelihood, Akaike information criterion (AIC), and Bayesian information criterion (BIC). The smaller values of AIC and BIC and the larger values of log likelihood indicate the better fit. The formulae for calculating the indicators are shown in Equations (8)–(10),

$$\text{LogLik} = ln(L); \qquad (8)$$

$$\text{AIC} = -2\ln(L) + 2k; \qquad (9)$$

$$BIC = -2\ln(L) + ln(n)k; \tag{10}$$

where $L$ is the likelihood function of the model, $ln(L)$ is the log likelihood function, $k$ is the number of model parameters, and $n$ is the number of observations.

**Table 7.** Factor-fitting method of multilevel nonlinear mixed-effects model.

| Model Number | Random Effect Number of Nesting Layers | Fixed Effect | Random Effect |
|:---:|:---:|:---:|:---:|
| 1 | | AC | |
| 2 | | SLC | |
| 3 | | EC | |
| 4 | Single level | AC + SLC | SC |
| 5 | | AC + EC | |
| 6 | | SC + EC | |
| 7 | | AC + SC + EC | |
| 8 | | SLC | |
| 9 | | EC | SC + AC |
| 10 | | SLC + EC | |
| 11 | | AC | |
| 12 | Two levels | EC | SC + SLC |
| 13 | | AC + EC | |
| 14 | | AC | |
| 15 | | SLC | SC + EC |
| 16 | | AC + SLC | |
| 17 | | EC | SC + AC + SLC |
| 18 | Three levels | SLC | SC + AC + EC |
| 19 | | AC | SC + SLC + EC |

In order to evaluate the prediction results of *Pinus densata* and compare the differences between observed and predicted values, the accuracy of the model needs to be tested. We selected the absolute mean relative error (AMRE), prediction precision ($P$), and root mean square error (RMSE) as the test indexes of the model. The specific calculation formulae are shown in Equations (11)–(13),

$$AMRE = \frac{1}{N} \sum_{i=1}^{N} \left| \frac{y_i - \hat{y}_i}{\hat{y}_i} \right| \times 100\% \tag{11}$$

$$P = \frac{1}{N} \sum_{i=1}^{N} \left( 1 - \left| \frac{y_i - \hat{y}_i}{\hat{y}_i} \right| \right) \times 100\% \tag{12}$$

$$RMSE = \sqrt{\frac{\sum_{i=1}^{N}(y_i - \hat{y}_i)^2}{N}} \tag{13}$$

where $y_i$ is the measured value, $\hat{y}_i$ is the model predicted value, and $N$ is the number of samples.

## 3. Results

### 3.1. Linear Mixed-Effects Model

We utilized NFI sample plot AGCS data and AGCS change data as dependent variables to fit the static and dynamic linear mixed-effects models. The texture eigenvalue and texture eigenvalue change were used as independent variables, while the soil class, aspect class, slope class, and elevation class were assigned as fixed and random effects. We then assessed the goodness of fit for 12 sets of static and dynamic models. The results revealed that models 3, 5, 8, and 10 for the static models, as well as models 2, 5, 7, and 10 for the dynamic models, performed better compared to other combinations. We calculated the estimation accuracy for each model, leading us to identify the best model for each type. For the static models, the best one included the elevation class as a fixed effect and the aspect class as a random

effect. Meanwhile, the best dynamic model incorporated slope rank as both a fixed and random effect. Both of these models combined mixed effects of aspect class and elevation class. Notably, the optimal dynamic model's estimation accuracy surpassed that of the static model by 30.42% (Table 8).

**Table 8.** Fitting accuracy of static linear mixed-effects model.

| Model Type | Model Number | Random Effect | Fixed Effect | Training Data | | Test Data | | |
|---|---|---|---|---|---|---|---|---|
| | | | | AMRE/% | RMSE /(t·hm$^{-2}$) | AMRE/% | RMSE /(t·hm$^{-2}$) | *P*/% |
| Static models | 3 | EC | AC | 74.005 | 37.323 | 56.219 | 38.360 | 43.781 |
| | 5 | AC | SLC | 56.367 | 57.132 | 52.193 | 54.316 | 47.807 |
| | 8 | AC | EC | 56.577 | 37.597 | 50.892 | 36.353 | 49.108 |
| | 10 | AC | SC | 56.924 | 37.719 | 51.716 | 36.778 | 48.284 |
| Dynamic models | 2 | SLC | AC | 17.507 | 23.062 | 20.472 | 28.583 | 79.528 |
| | 5 | AC | SLC | 18.703 | 23.809 | 22.444 | 30.413 | 77.556 |
| | 7 | SLC | EC | 18.395 | 23.894 | 23.278 | 32.254 | 76.722 |
| | 10 | AC | SC | 18.268 | 23.482 | 21.586 | 29.818 | 78.414 |

AC is aspect class, SLC is slope class, EC is elevation class and SC is soil class; same explanation as for the abbreviations in the later section.

### 3.2. Non-Parametric Model

We aimed to compare the differences between non-parametric and mixed-effects models. To achieve this, we fitted RF, GBRT, and K-NN using 117 data sets of static models AGCS data, with a 7:3 training-to-test data ratio. The modeling parameters and the maximum number of iterations were adjusted to obtain optimal results by fitting the model's multiple times.

For the RF model, the maximum number of iterations was set to 20, and the maximum depth of the decision tree was 10. For the GBRT model, the maximum number of iterations was 10, the maximum depth of the decision tree was 8, the learning rate was 0.05, and the subsampling ratio was 0.05. The K-NN model had a k-value of 10, with a uniform weighting method and Euclidean metric.

Similarly, we adjusted the modeling parameters and the maximum number of iterations for the dynamic model and obtained the best modeling parameters through multiple fittings. The RF model had a maximum number of iterations of 120, a maximum depth of the decision tree of 5, and the minimum number of samples for leaf nodes set to 2. The GBRT model parameters were set identically to those in the static GBRT. For the K-NN model, the k-value was 18, with a homogeneous weight calculation method and Euclidean metric.

The static model fitting results showed that the RF model had lower AMRE and RMSE values than both the GBRT and K-NN models. The best-fitting RF model demonstrated an AMRE of 23.499% and an RMSE of 9.190 t/hm$^2$. When testing the accuracy of each model using the test data, the RF model achieved an AMRE of 30.550%, an RMSE of 16.06 t/hm$^2$, and a prediction accuracy (P) of 69.45%. This accuracy was significantly higher than that of the GBRT and K-NN models and surpassed the static linear mixed-effects model's prediction accuracy.

In contrast, the dynamic non-parametric models' fitting results indicated that GBRT had lower AMRE and RMSE values than the RF and K-NN models, achieving the best fit. Specifically, the GBRT model had an AMRE of 17.267% and an RMSE of 11.23 t/hm$^2$. However, when evaluating the models using the test data, the RF model outperformed the others with an AMRE of 20.542%, an RMSE of 14.38 t/hm$^2$, and prediction accuracy (P) of 79.458%. This result was significantly higher than that of the the GBRT and K-NN models. Although the dynamic model demonstrated higher prediction accuracy than the static model (Table 9), the optimal non-parametric model displayed lower prediction accuracy compared to the dynamic linear mixed-effects model.

**Table 9.** Results of static non-parametric model fitting.

| Model Type | Model Form | Training Data | | Test Data | | |
|---|---|---|---|---|---|---|
| | | AMRE/% | RMSE/ (t·hm$^{-2}$) | AMRE/% | RMSE/ (t·hm$^{-2}$) | P/% |
| Static models | RF | 23.499 | 9.19 | 30.550 | 15.06 | 69.45 |
| | GBRT | 44.041 | 16.71 | 42.530 | 27.73 | 57.47 |
| | K-NN | 44.255 | 16.72 | 49.950 | 31.91 | 50.05 |
| Dynamic models | RF | 25.281 | 16.69 | 20.542 | 14.38 | 79.458 |
| | GBRT | 17.267 | 11.23 | 34.424 | 21.93 | 65.576 |
| | K-NN | 19.141 | 12.76 | 22.53 | 15.04 | 77.470 |

*3.3. Nonlinear Mixed-Effects Model*

3.3.1. Static Nonlinear Mixed-Effects Model

We initially fitted nine basic models, but four of them—the power function model, the logit model, the logistics model, and the forest stand growth model—failed to converge. This left us with five models that could be successfully fitted. We then fitted a total of 130 sets of nonlinear mixed-effects models, featuring different fixed and random effects as well as various levels of random-effects nesting.

From these 130 sets, we selected the 15 best-fit models under different stratified random effects for each base model. Due to the variation in the fixed-effects parameters modeled, it was necessary to examine the best-fit models for comparison across multiple levels of random-effects nesting within each of the five base models (Table 10).

**Table 10.** Goodness of fit of static nonlinear model.

| Base Model | Number of Random Effect Layers | Fixed Effect | Random Effect | LogLik | AIC | BIC |
|---|---|---|---|---|---|---|
| Natural logarithmic | | EC | | −409.98 | 827.96 | 837.58 |
| Polynomial | | AC | | −409.69 | 827.38 | 837.00 |
| Growth | Single level | AC + SLC | SC | −409.05 | 828.11 | 840.14 |
| Hyperbolic | | EC | | −409.76 | 827.53 | 837.15 |
| S-shaped curve | | AC + EC | | −408.26 | 826.51 | 838.54 |
| Natural logarithmic | | EC | | −399.35 | 806.70 | 816.33 |
| Polynomial | | SLC | | −399.29 | 806.59 | 816.21 |
| Growth | Two levels | SLC + EC | SC + AC | −398.09 | 806.19 | 818.22 |
| Hyperbolic | | SLC + EC | | −400.08 | 810.16 | 822.19 |
| S-shaped curve | | SLC + EC | | −398.09 | 806.17 | 818.21 |
| Natural logarithmic | | EC | | −393.06 | 794.13 | 803.76 |
| Polynomial | | EC | | −394.59 | 797.19 | 806.81 |
| Growth | Three levels | EC | SC + AC + SLC | −394.01 | 796.02 | 805.64 |
| Hyperbolic | | EC | | −392.66 | 793.32 | 802.95 |
| S-shaped curve | | EC | | −392.62 | 793.24 | 802.87 |

In the single-level random-effects model, the polynomial model, which utilized soil class as a random effect and slope class as a fixed effect, outperformed the other four models. In the two-level nested random-effects model, the polynomial model with nested random effects for soil class and slope class, along with slope class fixed effects, held fewer fixed-effects parameters than the S-curve model, resulting in lower BIC values.

Among the three classes of nested random effects models, the best was the S-curve model, which incorporated nested random effects for soil class, slope class, and slope class, and fixed effects for elevation class. To minimize uncertainty between the goodness of fit and estimation accuracy, we calculated the estimation accuracy for the 15 sets of models (Table 11). The results revealed that the nonlinear mixed-effects models performed relatively poorly with static AGCS data, showing polarization between estimation accuracy and goodness of fit. As the number of nested random effects increased, the goodness of fit improved, whereas the estimation accuracy declined. The three-level nested model achieved a maximum precision of 22.813%, while the highest precision was obtained for

the one-level random effects model, which closely resembled the static linear mixed-effects model. Therefore, when applying parametric models to single-period continuous inventory data proves challenging, non-parametric models become more suitable.

**Table 11.** Fitting accuracy of static nonlinear mixed-effects model.

| Base Model | Fix Effect | Random Effect | Number of Random Effect Layers | Training Data | | Test Data | | |
|---|---|---|---|---|---|---|---|---|
| | | | | AMRE/% | RMSE /(t·hm$^{-2}$) | AMRE/% | RMSE /(t·hm$^{-2}$) | P/% |
| Polynomial | AC | SC | Single level | 67.60 | 18.03 | 51.931 | 21.07 | 48.069 |
| Polynomial | SLC | SC + AC | Two levels | 51.53 | 18.41 | 54.503 | 27.73 | 45.497 |
| S-shaped curve | EC | SC + AC + SLC | Three levels | 49.02 | 18.05 | 77.187 | 20.86 | 22.813 |

3.3.2. Dynamic Nonlinear Mixed-Effects Model

We constructed a dynamic nonlinear mixed-effects model using 183 training data sets, with the amount of NFI AGCS variation in *Pinus densata* as the independent variable. Aspect class, slope class, and elevation class were combined in various forms as fixed effects, while soil, elevation, slope, and slope class were nested in a multilevel hierarchy as random effects.

In the static model, the presence of negative values in the texture information caused four inconvertible models to fail in converging. To address this issue, we normalized the variation in texture information in the dynamic model construction, ensuring no negative values in the independent variables. Despite this, the forest growth model and the logistics model still did not converge.

Ultimately, seven base models achieved convergence, resulting in 174 nonlinear mixed-effects models fitted with different fixed effects and graded random effects factors. Due to differing fixed-effects parameters, we individually selected the best-fit models for comparison among the multilevel random effects nested within the seven base models (Table 12). Each base model was screened for goodness of fit at different levels of random-effects nesting, producing a total of 21 sets of dynamic nonlinear mixed-effects models. The best-fit models for each base model at different levels of random effects nesting are presented. Our results show that the natural logit model with three levels of nesting has the best fit.

**Table 12.** Goodness of fit of dynamic nonlinear mixed-effects model.

| NOM | Base Model | Number of Random Effect Layers | Fixed Effect | Random Effect | LogLik | AIC | BIC |
|---|---|---|---|---|---|---|---|
| 1 | S-shaped curve | | AC | | −836.147 | 1688.293 | 1713.969 |
| 2 | Natural logarithmic function | | AC + SLC + EC | | −831.604 | 1683.201 | 1715.296 |
| 3 | Hyperbolic | | AC + SLC | | −834.728 | 1687.457 | 1716.342 |
| 4 | power | Single level | AC + SLC + EC | SC | −832.939 | 1685.879 | 1717.974 |
| 5 | Natural exponential | | AC | | −833.439 | 1682.878 | 1708.554 |
| 6 | Growth | | AC + SLC + EC | | −837.440 | 1694.879 | 1726.974 |
| 7 | Polynomial | | AC + SLC + EC | | −831.757 | 1683.513 | 1715.608 |
| 8 | S-shaped curve | | AC | SC + EC | −825.027 | 1666.055 | 1691.731 |
| 9 | Natural logarithmic function | | AC | SC + EC | −823.731 | 1663.462 | 1689.137 |
| 10 | Hyperbolic | | AC | SC + EC | −824.230 | 1664.459 | 1690.135 |
| 11 | power | Two levels | EC | SC + AC | −829.457 | 1674.913 | 1700.589 |
| 12 | Natural exponential | | AC | SC + EC | −824.695 | 1665.389 | 1691.065 |
| 13 | Growth | | AC | SC + EC | −826.096 | 1668.192 | 1693.868 |
| 14 | Polynomial | | AC + EC | SC + SLC | −822.201 | 1662.402 | 1691.288 |
| 15 | S-shaped curve | | SLC | SC + AC + EC | −817.199 | 1652.399 | 1681.284 |
| 16 | Natural logarithmic function | | EC | SC + AC + SLC | −817.080 | 1650.16 | 1675.836 |
| 17 | Hyperbolic | | EC | SC + AC + SLC | −817.227 | 1650.454 | 1676.13 |
| 18 | power | Three levels | EC | SC + AC + SLC | −817.644 | 1651.287 | 1676.963 |
| 19 | Natural exponential | | EC | SC + AC + SLC | −817.471 | 1650.942 | 1676.618 |
| 20 | Growth | | EC | SC + AC + SLC | −818.443 | 1652.887 | 1678.563 |
| 21 | Polynomial | | None | SC + AC + EC | −823.823 | 1661.646 | 1684.113 |

None: no selection fixed effects.

We calculated the fit and prediction accuracies for the 21 groups of models and selected the models with the highest prediction accuracy at each level of random-effects nesting for comparison (Table 13). The results indicated that Model 6 had a prediction accuracy of 80.265% for the single-level random effect, while Model 13 achieved an accuracy of 80.433% for the two-level nested random effect, and Model 20 reached 76.202% for the three-level nested random effect. Notably, all three models were growth models, and among them, Model 13 outperformed the other 20 model groups in all accuracy measures for the two-level nested random-effects growth model.

**Table 13.** Prediction accuracy of dynamic nonlinear mixed-effects model.

| Number of Models | Base Model | Fixed Effect | Random Effect | Number of Random Effect Layers | Training Data | | Test Data | | |
|---|---|---|---|---|---|---|---|---|---|
| | | | | | AMRE/% | RMSE /(t·hm$^{-2}$) | AMRE/% | RMSE /(t·hm$^{-2}$) | P/% |
| 6 | Growth | AC + SLC + EC | SC | Single level | 17.834 | 11.88 | 19.735 | 13.65 | 80.265 |
| 13 | Growth | AC | SC + EC | Two levels | 17.646 | 11.64 | 19.557 | 13.53 | 80.443 |
| 20 | Growth | EC | SC + AC + SLC | Three levels | 18.454 | 12.28 | 23.798 | 16.40 | 76.202 |

To identify the differences between various random effects after nesting, we should simultaneously calculate the random-effects covariance parameters for the models mentioned above. Since the number of nested levels will result in different matrix ranges for different random effects, we need to design three random-effects covariance moments ($1 \times 1$, $2 \times 2$, and $3 \times 3$). By setting the covariance matrix uniformly as a diagonal matrix, we can compute the final random-effects parameters of the model by fitting the entire parameter set into the matrix. We find that the variance–covariance of the random effects decreases as the level of nesting of the random effects increases, indicating that the correlation between the random effects is gradually increasing. This implies that there is some correlation between the soil and terrain in which the forests are located, and the trend of forest carbon stock changes in the same soil conditions and terrain may be similar (Table 14).

$$D_1 = \begin{bmatrix} U_1 U_2 \\ U_2 U_1 \end{bmatrix} \tag{14}$$

$$D_2 = \begin{bmatrix} U_3 & U_2 & U_1 \\ U_2 & U_1 & U_3 \\ U_1 & U_3 & U_2 \end{bmatrix} \tag{15}$$

**Table 14.** Random-effects parameter calculation results.

| Number of Models | Base Model | Number of Random Effect Layers | Variance–Covariance Matrix | Random Effect |
|---|---|---|---|---|
| 6 | Growth | Single level | 474.093 | 0.014 |
| 13 | Growth | Two levels | 414.264 | 0.023 |
| 20 | Growth | Three levels | 336.000 | 0.032 |

In the equations above, $U_1$, $U_2$, and $U_3$ are random-effects parameters, $D_1$ is a two-level nested covariance matrix, and $D_2$ is a three-level nested covariance matrix.

### 3.4. Parameter Results of Dynamic Nonlinear Mixed-Effects Model

In our final analysis, we compared the prediction accuracy of static and dynamic models together. Among the static models, the prediction accuracy of the RF model was 69.45%, with an RMSE of 15.06 t/hm$^2$ and an AMRE of 30.55%—all noticeably lower than the other four static models. For the dynamic models, the nonlinear effects model yielded a significantly higher prediction accuracy of 80.44% compared to the other four models within its category and the five static models. Although it had an RMSE of 13.53 t/hm$^{-2}$ and an AMRE of 19.56%, these values were still notably lower than those of the other nine

models (Figure 7). Consequently, we adopted the dynamic nonlinear mixed-effects model as the carbon stock inversion model in our study.

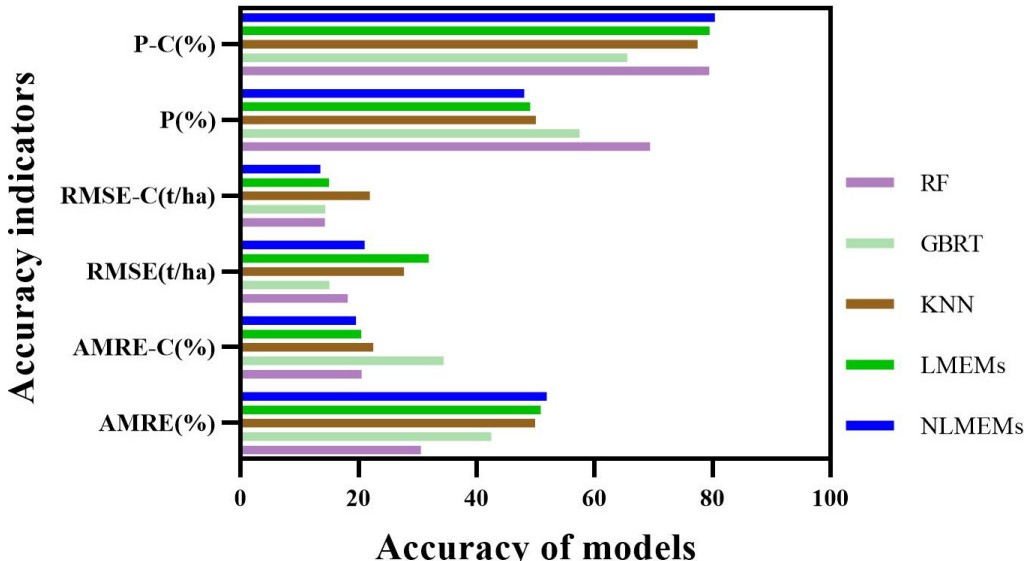

**Figure 7.** Estimation accuracy comparison. P-C, RMSE-C, and AMRE-C are indicators of the estimation accuracy of the dynamic model.

From the calculations above, we derived the optimal dynamic nonlinear mixed-effects model's basic constitutive form and parameter composition. The base model is a growth function, with fixed effects stemming from the grades in the aspect. The fixed variables include R19B5EN, R15B5EN, R17B5ME, R15B5ME, R19B5ME, and R17B5DS. Additionally, the random effects consist of soil grades nested at two levels with elevation grades (Table 15).

**Table 15.** Parameters of nonlinear mixed-effects model of above-ground carbon stock of *Pinus densata* Mast.

| Model Parameters | Parameter | Parameter Value |
|---|---|---|
| Constant term | B0 | 2.601009 |
| Fixed variable | B1(R19B5EN)<br>B2(R15B5EN)<br>B3(R17B5ME)<br>B4(R15B5ME)<br>B5(R19B5ME)<br>B6(R17B5DS) | 0.7608801<br>−0.3712685<br>1.461169<br>−1.017067<br>−0.4221513<br>−0.3101006 |
| Fixed effects | B7(Aspect Class) | −0.0260525 |
| Random effects | U (Elevation Class × Soil Class) | 0.0225032 |
| Variance–covariance matrix | D | 414.2642 |
| Random effects error | $e_{ij}$ | 0.0108976 |
| Log likelihood | −826.09595 | |
| AIC | 1668.192 | |
| BIC | 1693.868 | |

### 3.5. Carbon Stock Calculation Based on Mixed-Effects Model

We employed an optimal nonlinear mixed-effects model to calculate the above-ground carbon stocks (AGCS) for *Pinus densata* in the Shangri-La region. This enabled us to determine the AGCS of *Pinus densata* in Shangri-La for the period between 1987 and 2017

(Table 16). Subsequently, we produced a map illustrating the spatial distribution of carbon stocks (Figure 8).

**Table 16.** Comparison of model estimation accuracy.

| Year | *Pinus densata* Area (hm$^{-2}$) | AGB (t) | Carbon Storage (t) | Unit Area Carbon Stocks (t·hm$^{-2}$) |
|---|---|---|---|---|
| 1987 | 219,761.820 | 8,586,720.848 | 4,301,947.145 | 19.575 |
| 1992 | 171,567.720 | 6,910,726.316 | 3,462,273.884 | 20.180 |
| 1997 | 170,583.660 | 6,914,485.838 | 3,464,157.405 | 20.308 |
| 2002 | 170,583.660 | 7,540,668.431 | 3,777,874.884 | 22.147 |
| 2007 | 174,182.490 | 7,610,497.882 | 3,812,859.439 | 21.890 |
| 2012 | 174,216.510 | 8,039,281.133 | 4,027,679.848 | 23.119 |
| 2017 | 184,806.990 | 9,337,490.337 | 4,678,082.659 | 25.313 |

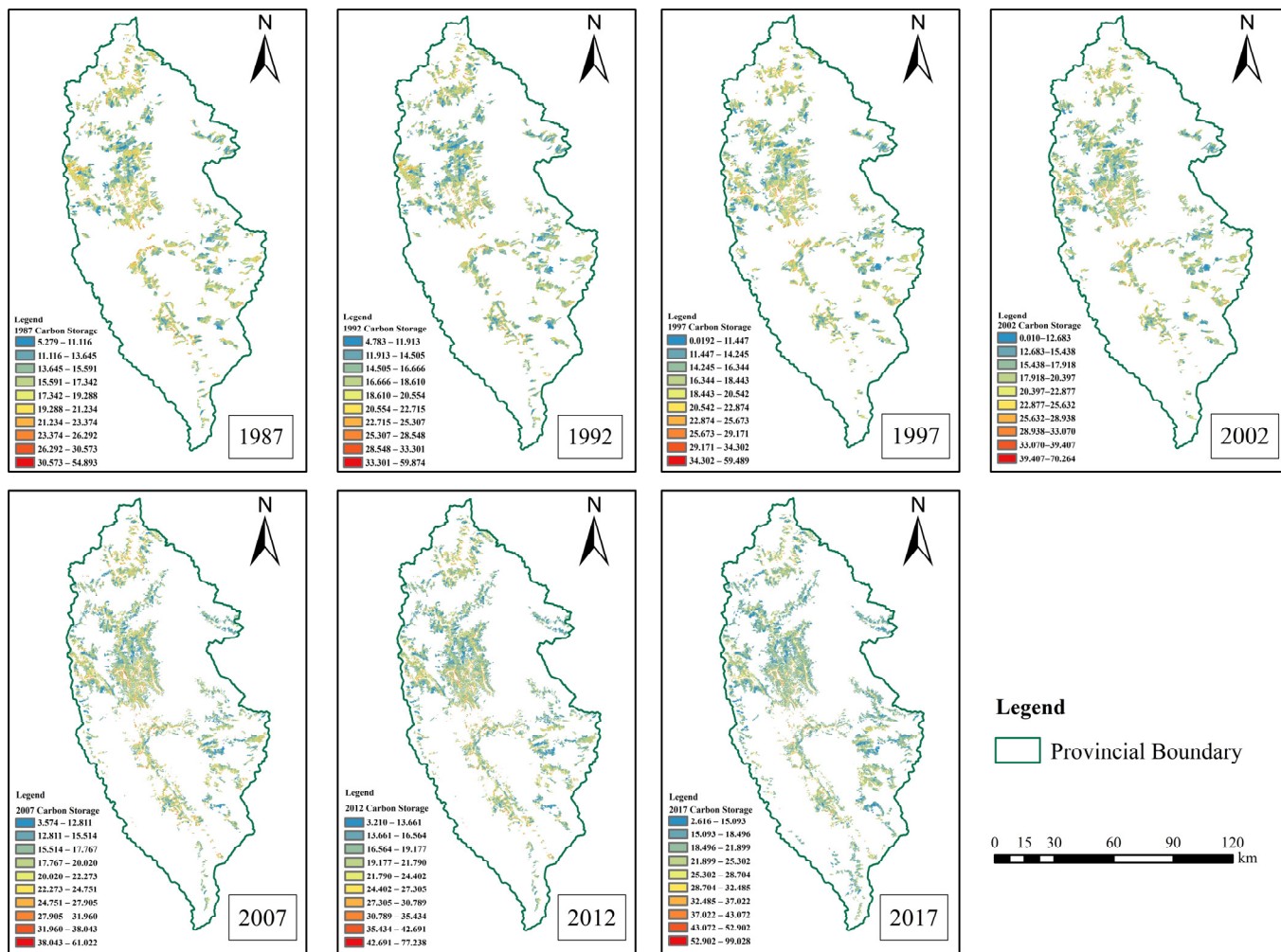

**Figure 8.** Thirty-year spatial and temporal distribution of carbon stock.

## 4. Discussion

### 4.1. Model Accuracy Issues

Significant differences exist between the dynamic forest inventory data and the artificial single-period sample plot selection. Manual single-period sample sites are often selected using selective sampling methods, which yield smaller data variability due to samples chosen for similar situations. This, combined with human measurement error, may result in reduced sample representativeness and systematic bias toward overestima-

tion [75]. On the other hand, continuous forest surveys are conducted every five years using systematic sampling, where sample units are arranged according to a predetermined pattern [76,77]. This approach ensures more objective data representation and higher NFI data variability. In our study, the largest sample plot AGCS was still over 80 times larger than the smallest sample plot AGCS after removing outliers using the triple standard deviation method. This study utilized AGCS change modeling, which improved the estimation accuracy by 8.11%–32.37%, with the greatest improvement observed in NLMEMs. This increased accuracy may be due to a larger sample size, which often fails to converge when fitting models with static year data [78]. Modeling with data from different year variations not only increases the sample size but also effectively improves the model fit. Consequently, the study of mixed-effects models should consider the impact of sample size on fit to enhance model accuracy [79]. The AGCS change model is more suitable for remote sensing estimates of AGCSs of conifers over long time series.

In this study, two types of models were constructed: non-parametric and parametric models. These models demonstrate differing estimation accuracies under various modeling scenarios. Among the models for the static estimation of AGCS, the Random Forest model has the highest prediction accuracy, similar to the results of Zhang et al. [80]. Non-parametric models are advantageous for their lesser dependence on the overall distribution of the sample, but lack a specific model form and may be uncertain when applied to other regions or periods. Bayesian non-parametric methods are often employed to quantify model uncertainties in such cases [81]. Trees are typical of spatial point pattern data, and some studies indicate that parametric models may fit spatial point pattern data better than non-parametric models, provided the assumptions of the parametric model hold [82,83]. The linear and nonlinear mixed-effects models constructed in this study achieved high predictive accuracy for both single-period and long-time-series AGCS estimates. While the parametric model has a specific form that can be potentially applied to other regions, further modeling and testing are required to determine its suitability.

### 4.2. Selection of Modeling Variables

Uncertainty about carbon sinks in terrestrial ecosystems has long been a challenge in addressing climate change, primarily due to the difficulty in handling the spatial heterogeneity of carbon sink estimates. Both fixed methodological systems and parameter default values present challenges for forest carbon sink estimation. To address this issue, we should build on existing estimation approaches and investigate methods to reduce the uncertainty associated with the spatial heterogeneity of forest carbon sinks. DEM and DSM data from different years provide valuable global surface and topography information [84], which, when matched to study plots, enables quick access to monitoring sample site information such as elevation, slope orientation, and gradient. By introducing DEM topographic information at a 30m spatial resolution to inventory methods for estimating carbon stocks, we were able to achieve spatial data analysis, matching the pixel values of Landsat imagery. However, the uncertainty of DEM is often underestimated, and further exploration of topographical uncertainty across temporal and spatial scales is needed [85].

Our results are consistent with the Introduction, suggesting a strong correlation between dynamic forest AGCS changes and texture information derived from medium-resolution remote sensing imagery. Although various vegetation indices and complex band ratios were extracted, texture information—represented by DN values reflecting the grey-scale value of each pixel [86], and the entropy, mean, and dissimilarity of texture information—displayed the strongest correlation with AGB changes. In this study, we constructed a linear and nonlinear mixed-effects estimation model of the AGB of *Pinus densata* based on remotely sensed eigenvalues. Results from another study report the AMRE of a mixed-effects model for estimating *Pinus densata* AGCS in single-period anthropogenic survey sample sites to be 31.52%, with a prediction accuracy of 77.83% [87]. Although the selection of areas as mixed effects reflects spatial heterogeneity, using areas as random effects only may not accurately represent the specific topographic effects in different regions.

In our previous study, we constructed a nonlinear mixed-effects model using elevation class and sample site location as fixed effects and slope class as random effects to estimate the AGB of *Pinus densata*. After optimizing the model's nonlinear form, the best model achieved an AMRE of 15.64% and a prediction accuracy of 84.35% [88]. The accuracy of AGCS estimates is influenced by differences in climate and species richness along the altitudinal gradient [89], as well as aspect orientation, which affects vegetation distribution, light duration [90,91], and reflectance. Greater attention should be paid to aspect orientation in remote sensing assessments [92]. Additionally, the irregular spatial distribution of soils on different slopes and aspects demonstrates an extremely strong correlation [93]. No studies have yet employed mixed-effects models in estimating carbon stocks at *Pinus densata* sample sites using long remote sensing time series. Our study achieved similarly good results when using anterior–posterior changes in topography as a mixed-effects factor to estimate the dynamics of above-ground forest carbon stocks. These results further indicate the effectiveness of topography in improving the estimation of spatial variability through remote sensing.

### 4.3. Multilevel Hierarchical Nesting Studies

In this study, we examine nonlinear mixed-effects models, which are divided into single-level random effects models and multilevel nested random effects models. Multilevel nested models represent the broader subject, with single-level random effects defaulting to a variance–covariance matrix as the unit matrix, where all random effects have the same variance. The variance–covariance matrix in multilevel random effects is determined by the number of levels of random effects; each nested level consists of that random effect factor with different levels of sample individuals in other groups [94].

We use continuous inventory data of forest resources, which meet the criteria for repeated measurement data. Multilevel nested random effects offer a more flexible covariance structure than traditional fixed-effects models, taking into account the variation in individuals in the longitudinal data [95,96]. This approach considers not only the AGCS variation within the fixed sample plots but also how changes in topography and soil thickness influence AGCS variation. The uncertainty and impact of these stochastic factors on AGCS estimates are reflected by specific parameters that modify the estimation accuracy.

Our study involves a maximum of three nested levels of random effects; however, the accuracy of the model does not increase with the number of nested levels. When there are three nested levels, the model's prediction accuracy is lower than that of a two-level nested model. Since NFI sample point data typically have temporal and spatial dimensions with individual heterogeneity intercepts, we utilize topography and soil as random intercepts. We assume that the heterogeneity intercepts of individual sample sites are related to the remote sensing images' spectral and textural information. Introducing them as dummy variables in the regression can eliminate random individual differences and improve the estimation effect. In future studies, the number of random-effects nested layers and their optimal combination should be carefully considered, as should the choice of random effects.

Currently, researchers constructing mixed-effects estimation models for forest AGCS typically combine individual wood data measured in sample plots, stand factors accounting for environmental factors, and remote sensing data for modeling [97,98]. Our study uses only remotely sensed eigenvalues and topographic factors for model construction, featuring a single modeling condition without meteorological factors. Various studies have shown that mixed-effects models incorporating climatic factors can effectively reflect environmental gradients' effects on stand distribution and growth [99,100]. Combining climate factors with remote sensing-based carbon stock estimates may be considered in future studies to better analyze environmental drivers of carbon stock change.

Our study's observation samples are relatively small, with the random-effects parameters being relatively simple, including single-level random-effects parameters and multilevel random-effects parameters. The random effect variance–covariance structure consists primarily of a diagonal matrix with a relatively simple matrix structure. We em-

ploy AIC, BIC, and likelihood ratios to determine the random effects' effect using different combinations. Lower AIC and BIC values indicate better fit, mainly taking advantage of the likelihood ratio test in mixed-effects models [101].

An interesting issue concerns the discrepancy between the goodness of fit and the accuracy of fit of the mixed-effects model. Our results show that the growth model has the highest estimation accuracy, but its AIC, BIC, and Loglik values are not optimal. This outcome may result from different base models' forms, where AIC, BIC, and Loglik values determine the optimal model within the same base model but cannot be used for comparisons between different base models to avoid excessive variance and overfitting risks [102]. The most suitable solution is to test each base model's optimal parameter combination to reduce method-introduced uncertainty.

In conclusion, this study selects only nine nonlinear base models and does not adjust the model form. While this somewhat avoids overfitting, we may not have found the optimal model form. Further research should use more nonlinear models with higher AGCS or carbon stock correlations and adjust the model parameters to find the most suitable model for estimating *Pinus densata* AGCS.

*4.4. Carbon Stock Change Drivers and Uncertainty*

Uncertainty analysis is crucial in forest carbon stock measurement, with numerous sources of uncertainty, including systematic biases like data errors, inaccuracies in models or methods, and random errors arising from variations between measurement samples [12,103]. While data errors are generally the most influential, this study focuses on distinguishing between uncertainties in different methods rather than examining data measurement accuracy or sample representativeness. Furthermore, the *Pinus densata* anisotropic growth equation used here is based on researchers' actual sample plots instead of NFI fixed sample points, possibly leading to inaccuracies in reflecting local tree samples in other locations [104].

Shangri-La's complex, undulating terrain is dominated by high-mountain pine forests that are predominantly natural. These natural forests have more intricate ecosystems compared to planted forests and are subject to policy-driven changes. From 1987 to 1992, carbon stocks in *Pinus densata* natural forests decreased significantly by 692,185.741 t, and the area of these forests shrunk by 48,194.1 $hm^{-2}$ due to land use changes. While the decline slowed after 1992, carbon stocks began to increase after 2000, primarily driven by policy. Yunnan Province was included in the pilot project for the first phase of natural forest protection [105], and a policy of returning farmland to forests was implemented, mitigating the decline in *Pinus densata* natural forest areas. However, because of local practices, some irregular tree harvests persisted [106]. As a result, between 2002 and 2007, the area of pine forests increased, but carbon stock per unit area decreased by 0.212 t.

Although the moratorium on commercial logging in natural forests improved harvesting practices, it may also contribute to forest degradation due to inadequate scientific and effective human management [107]. Between 2012 and 2017, while natural forest areas expanded by 10,590.48 hectares, the carbon stock per unit area grew by only 2.194 $t \cdot hm^{-2}$. Given the tight timeframe to achieve climate targets, natural forest management must prioritize high-carbon-density areas. The scientific and sustainable management of large natural forest areas can enhance forest quality and significantly contribute to climate goals (Table 16).

**5. Conclusions**

Inventory methods are essential for bottom-up regional carbon stock estimation. To achieve accurate estimates, addressing the spatial heterogeneity of data and improving the methods' generalizability is crucial. Methods for analyzing space-sensing data, such as those using Landsat TM and OLI time-series imagery combined with NFI data, enhance regional carbon stock estimates. The resulting nonlinear mixed-effects carbon stock estimation models improve the accuracy of the estimates but reveal certain challenges.

The accuracy of remote sensing models for regional carbon stock estimation can be improved by leveraging the strong correlation between static and dynamic carbon stock data, texture information from Landsat imagery, and the variability of texture information. The non-parametric model demonstrated the highest accuracy in static AGCS estimation, while the parametric model excelled in estimating dynamically changing AGCS variables. Overall, the dynamic models outperformed the static ones. Incorporating both carbon stock change and remote sensing information change as modeling variables is advisable for the remote sensing estimation of spatial and temporal changes in above-ground forest carbon stocks.

The number of nested random effects of nonlinear mixed effects significantly impacts the prediction accuracy of the model, with the two-level hierarchical nested random effects providing the highest accuracy. Including topography and soils reduced the spatial heterogeneity of the inventory method but did not guarantee a direct link between the goodness-of-fit index and estimation accuracy. Thus, testing the accuracy of each model estimate is essential in constructing mixed-effect models.

Our method estimates the temporal and spatial variability of carbon stocks in *Pinus densata* in Shangri-La over 30 years more accurately, identifies temporal trends and spatial distributions of carbon stocks, and offers the parameter form of the best nonlinear mixed-effects model. However, due to limited *Pinus densata* forest sampling site data from other study areas, further research and testing are needed to assess generalizability across regions.

In conclusion, this research generates a thematic map that illustrates the spatial and temporal variability of carbon stocks within the Shangri-La region. It introduces a method leveraging remote sensing to refine the parameterization of terrestrial carbon stocks, particularly in areas of high altitude and complex terrain. As the advantages of remote sensing for estimating carbon stocks become increasingly apparent, this approach marks a significant step towards more effective and informed strategies for meeting local and national climate objectives.

**Author Contributions:** Conceptualization, D.H. and S.C.; methodology, J.Z.; software, R.B.; formal analysis, D.H.; data curation, D.H.; writing—original draft preparation, D.H.; writing—review and editing, S.C. and D.X.; visualization, Y.L. and S.W.; project administration, S.C.; funding acquisition, S.C. All authors have read and agreed to the published version of the manuscript.

**Funding:** This research was funded by the Cooperative Forestry Science and Technology Project of Zhejiang Provincial Academy (Study on carbon sequestration and sink enhancement technology of forest ecosystem in Zhejiang Province and the realization path of carbon sink value, No. 2023SY02); Research on Key Technologies and Paths for Realizing the Value of Ecological Products under the Special Funds for Basic Scientific Research of Institutions of Public Welfare at Central Level (CAFYBB2022MC001); and the National Natural Science Foundation of China (Nos. 31860207 and 32260390).

**Data Availability Statement:** The data presented in this study are available on request from the corresponding author. The data are not publicly available due to the confidentiality of the NFI data set.

**Conflicts of Interest:** The authors declare no conflicts of interest.

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
