# Peer review of "Improving Pinus densata Carbon Stock Estimations through Remote Sensing in Shangri-La: A Nonlinear Mixed-Effects Model Integrating Soil Thickness and Topographic Variables"

_forests, doi:10.3390/f15020394_

Round 1
Reviewer 1 Report
Comments and Suggestions for Authors
This study estimates soil and topography-accounting models for estimating forest carbon stocks using remote sensing techniques. Overall, the article is technically sound; however, there are several questions and remarks regarding the manuscript.
The data for the 20 fixed sample sites from China's Continuous National Forest Inventory were used in this study (Lines 181-182). Given the extent covered by Pinus densata forests in the region, a forest inventory was conducted using one sample site every 92.4 square kilometres. How accurate is it to calculate carbon stocks using such extrapolated data?
Of the 32 fixed sample sites that were initially identified for the study, only 20 were selected for further investigation. Why were twelve sites left out of the research?
“In this study, we developed a remote sensing method…” (Lines 21-22). You developed a method for analysing space sensing data, not a remote sensing method. It is necessary to be correct in the wording.
“…or 16.18% of the city's total land area…” (Line 168) A city?
Table 2. Use official band names, please. (https://www.usgs.gov/landsat-missions/landsat-5 (see: Thematic Mapper (TM)))
Why were various months of the year chosen for the research to collect the initial satellite sensing data (Table 1)? Could seasonality (the time at which the space image was created) impact the accuracy of your research?
“Resampling the image to a resolution of 30 m × 30 m…” (Line 211). With the exception of Band 6, which has a resolution of 120 m, the initial resolution for Landsat 5 and Landsat 8 is 30 m. Explain why resampling was conducted.
The predominant species found in the area include Quercus semicarpifolia, Pinus yunnanensis, Pinus densata, Picea asperata, and Abies fabri. You have investigated Pinus densata forests. Using all the remote sensing data, how did you identify the pine forests?
Reviewer 2 Report
Comments and Suggestions for Authors
Dear Authors,
2. Materials and Methods
2.1. study area
in my opinion, the authors should include basic information about the soils of the study area in this chapter. The authors should specify which soil types are represented for the study area, according to the FAO-WRB classification system. In addition, the authors should provide information on the texture classes with which the study area was characterised. In general, in the paper, the authors have insufficiently presented soil information for the study area.
The impact of the soil on organic carbon stocks is fundamental and should therefore be properly pointed out in the work
Reviewer 3 Report
Comments and Suggestions for Authors
Overall a very well-written paper which is clearly written, logically structured, and appears to be a sound piece of research. The research is accessible and the purpose clear. At this stage, I would really only suggest that the authors undertake a review of the English and make some minor edits, perhaps even just using something like Grammarly.
The only thing that crossed my mind is perhaps more of a question than anything else relates to the comparison of the assessments using the proposed methods with the ground-based/other methods...... how do you know how accurate the ground-based methods are for comparison? I recall something I read recently which suggested that a lot of the carbon sequestration assessments for many woodland/forest areas are a long way out now based on how the assessments were made and how these areas have changed over time... which is partly the thinking behind this research...
Comments on the Quality of English LanguageVery good and only a need to undertake minor corrections.
Reviewer 4 Report
Comments and Suggestions for Authors
I suggest requesting a major overhaul of the manuscript. In the present form the text is not ready for submission. The text contains many typographical errors that need to be fixed before the text can undergo review.
Technical question: the authors could be specific about the spatial extent of the validity of their model.
Can a similar model be developed for any region?
Typography:
There are many missing spaces or extra commas before full stops and commas, and parentheses.
There are several misplaced full stops.
Please check superscripts.
The plant names are inconsistent, partially italics, partially starting with capitals, and partially incomplete.
A thorough check is required before submission.
Comments on the Quality of English Language
see comments above.
Round 2
Reviewer 1 Report
Comments and Suggestions for Authors
All of the issues mentioned during the initial round of review were thoroughly addressed. The manuscript has been carefully revised.
Reviewer 4 Report
Comments and Suggestions for Authors
The revised manuscript is a significant improvement as compared to the previous version.
I recommend the publication.